# Comparison of Two Software Packages for Perfusion Imaging: Ischemic Core and Penumbra Estimation and Patient Triage in Acute Ischemic Stroke

**DOI:** 10.3390/cells11162547

**Published:** 2022-08-16

**Authors:** Xiang Zhou, Yashi Nan, Jieyang Ju, Jingyu Zhou, Huanhui Xiao, Silun Wang

**Affiliations:** 1Department of Radiology, Tongji Hospital, School of Medicine, Tongji University, 389 Xincun Rd., Shanghai 200065, China; 2YIWEI Medical Technology Co., Ltd., Room 1001, MAI KE LONG Building, Shenzhen 518000, China; 3The Second Affiliated Hospital of Nanjing Medical University, 121 Jiangjiayuan Rd., Nanjing 210011, China

**Keywords:** RealNow, RAPID, ischemic core, penumbra, acute ischemic stroke

## Abstract

**Purpose:** Automated postprocessing packages have been developed for managing acute ischemic stroke (AIS). These packages identify ischemic core and penumbra using either computed tomographic perfusion imaging (CTP) data or magnetic resonance imaging (MRI) data. Measurements of abnormal tissues and treatment decisions derived from different vendors can vary. The purpose of this study is to investigate the agreement of volumetric and decision-making outcomes derived from two software packages. **Methods:** A total of 594 AIS patients (174 underwent CTP and 420 underwent MRI) were included. Imaging data were accordingly postprocessed by two software packages: RAPID and RealNow. Volumetric outputs were compared between packages by performing intraclass correlation coefficient (ICC), Wilcoxon paired test and Bland–Altman analysis. Concordance of selecting patients eligible for mechanical thrombectomy (MT) was assessed based on neuroimaging criteria proposed in DEFUSE3. **Results:** In the group with CTP data, mean ischemic core volume (ICV)/penumbral volume (PV) was 14.9/81.1 mL via RAPID and 12.6/83.2 mL via RealNow. Meanwhile, in the MRI group, mean ICV/PV were 52.4/68.4 mL and 48.9/61.6 mL via RAPID and RealNow, respectively. Reliability, which was measured by ICC of ICV and PV in CTP and MRI groups, ranged from 0.87 to 0.99. The bias remained small between measurements (CTP ICV: 0.89 mL, CTP PV: −2 mL, MRI ICV: 3.5 mL and MRI PV: 6.8 mL). In comparison with CTP ICV with follow-up DWI, the ICC was 0.92 and 0.94 for RAPID and Realnow, respectively. The bias remained small between CTP ICV and follow-up DWI measurements (Rapid: −4.65 mL, RealNow: −3.65 mL). Wilcoxon paired test showed no significant difference between measurements. The results of patient triage were concordant in 159/174 cases (91%, ICC: 0.90) for CTP and 400/420 cases (95%, ICC: 0.93) for MRI. **Conclusion:** The CTP ICV derived from RealNow was more accurate than RAPID. The similarity in volumetric measurement between packages did not necessarily relate to equivalent patient triage. In this study, RealNow showed excellent agreement with RAPID in measuring ICV and PV as well as patient triage.

## 1. Introduction

Computed tomographic perfusion (CTP) as well as MRI techniques of diffusion-weighted imaging (DWI) and perfusion-weighted imaging (PWI) have been used for the management of acute ischemic stroke (AIS) [1,2,3,4]. Software packages that can automatically postprocess perfusion imaging were developed to facilitate radiologists and medical doctors in making timely and precise clinical decisions about AIS treatment [1]. These packages compute perfusion maps including relative cerebral blood volume (rCBV), relative cerebral blood flow (rCBF), mean transit time (MTT) and time to maximum peak (T_max_) [5]. These perfusion maps represent hemodynamic properties, and they are used to determine ischemic core and penumbra regions [6].

The DAWN and DEFUSE3 trials have confirmed that mechanical thrombectomy (MT) can improve clinical outcomes for AIS patients selected with either CTP or MRI [7,8]. These trials were based on postprocessing results derived from RAPID (iSchemaView, Menlo Park, CA, USA) to triage patients. Inclusion criteria for MT candidacy were proposed in DEFUSE3 as ischemic core volume (ICV) < 70 mL, mismatch ratio > 1.8 and mismatch volume > 15 mL. In clinical centers relying on software packages other than RAPID, extrapolation of the inclusion criteria needs to be validated [9].

Theoretically, similar measurements of ischemic core and penumbra show equivalent performance among packages [10]. However, different biases (i.e., either overestimation or underestimation) in volumes of ischemic core and penumbra can act associatively and affect patient triage differently [11]. For instance, a patient with a slightly underestimated penumbra and an overestimated ischemic core is likely to be excluded from MT improperly because of a misleading mismatch profile [12]. Prior studies evaluated the differences in measurements among packages, but the impact of volumetric differences on making medical decisions was not addressed at the patient level.

This study aimed to assess the differences between two packages: RAPID and RealNow (Dr. Brain, China), in terms of volumetric estimation and patient stratification based on CTP and MRI data. Similar to RAPID, RealNow is an automated postprocessing package for perfusion imaging, and it has been applied in some clinical centers with no access to RAPID.

## 2. Materials and Methods

### 2.1. Study Design and Patients

This was a multicenter and retrospective study. We reviewed AIS patients admitted to the First Affiliated Hospital of Nanjing Medical University, the Second Affiliated Hospital of Nanjing Medical University and Tongji Hospital of Tongji University between December 2015 and July 2021. We retrieved clinical and radiological data, including patient gender, age, site of vascular occlusion, CTP, or MR images. We included patients who met the following criteria: (1) CTP or MRI should be caught on arrival at the stroke center within 6–24 h from symptoms onset, (2) age ≥ 18 years and (3) AIS caused by intracranial large artery occlusion. We excluded cases that (1) had poor scan quality, (2) did not follow the acquisition protocol recommended by software packages and (3) failed in the automated postprocessing method. The study was approved by the Ethics Committee of Tongji Hospital (approval number: K-2020-021). Written informed consent for participation was not required for this study in accordance with the national legislation and the institutional requirements.

### 2.2. Image Acquisition and Postprocessing

CTP data were acquired on either Siemens or Toshiba CT scanner, while MRI data were acquired on Philips MR scanner. The scan protocols are summarized in Appendix A.

Imaging data were processed using CT or MR postprocessing modules provided by two fully automated software packages: RAPID (referred to as package A) and RealNow (referred to as package B). The main postprocessing steps for CTP and MRI data were illustrated in Figure 1. Details of the algorithms were identical to the vendors, but the necessary steps were consistent. For CTP postprocess, both the ischemic core and the penumbra were determined using a time-series data set; for MRI postprocess, the ischemic core was determined using DWI, and the penumbra was determined using PWI. Shortly, imaging data were preprocessed for motion correction, smoothing and skull-stripping, after which an arterial input function (AIF) that reflected the contrast agent concentration versus time was determined. Then, perfusion parameters were computed using deconvolution algorithms. Both packages calculated perfusion parameters including T_max_, rCBV, rCBF and MTT with definitions as follows:$$\mathrm{rCBV}=100\cdot \frac{k_{AV}}{\rho }\frac{1-H_{SV}}{1-H_{LV}}\frac{\int c_{t}\left(t\right)dt}{\int c_{a}\left(t\right)dt},\;[\mathrm{mL}/100\;g]$$
$$\mathrm{rCBF}=100\cdot 60\cdot \frac{k_{AV}}{\rho }\left[max\left(r\left(t\right)\right)\right],\;[\mathrm{mL}/100\;g/\min]$$
$$\mathrm{MTT}=\frac{60\cdot \mathrm{CBV}}{\mathrm{CBF}},\;[s]$$
$$T_{\max}=argmax\left[r\left(t\right)\right],\;[s]$$
where $c_{t}\left(t\right)$ is perfusion image signal, $c_{a}\left(t\right)$ is arterial inflow function, r(t) is tissue property, ρ is density of brain and $H_{SV}$, $H_{LV}$ and $k_{AV}$ are scaling factors.

More details about stages of RealNow software processing were introduced in Appendix A.

Package A is FDA approved for clinical use. It identified abnormal tissues with thresholds: ischemic core with a threshold as rCBF less than 30% for CTP or apparent diffusion coefficient (ADC) less than 620 × 10^−6^ mm^2^/s for MRI and penumbra with a threshold as T_max_ greater than 6 s. These thresholds were chosen based on previous published receiver–operator-curve (ROC) analyses as the one that maximized the accuracy of lesion identification. However, these previous validated thresholds may not apply in the setting of early and complete reperfusion [13].

The default thresholds for identifying abnormalities in package B were the same as those adopted in package A. Additionally, package B provides a user interface online that allows the thresholds for rCBF, ADC and T_max_ to be self-defined, and thereby radiologists can evaluate perfusion abnormalities dynamically (Figure 2) and define lesion areas accurately with optimal thresholds chosen specifically under different circumstances for a heterogeneous population of AIS patients. To compare package A and B, we selected the default thresholds for ischemic core and penumbra estimation.

In order to validate the accuracy of CTP ischemic core volume (ICV), this study included 24 h follow-up DWI with manual lesion labeling as ground truth. AIS lesions with high signal on DWI should be referenced to the corresponding areas on the apparent diffusion coefficient maps to avoid selecting regions of T2 shine through. The perceived boundary of AIS lesions on DWI were manually delineated by one 8-year-experienced neuroradiologist using the open-source software ITK-Snap (version 3.8.0) (http://www.itksnap.org) (accessed on 28 June 2022). The volume of AIS lesions on DWI was calculated from each labeling slice.

### 2.3. Data Analysis

Statistical analysis was performed with MedCalc 20.019 (MedCalc software, Ostend, Belgium). Continuous variables were reported using mean (standard deviation, SD) and median (interquartile range, IQR); discrete variables were reported using counts (percentages).

The volumes of ischemic core and penumbra obtained from packages were compared. We performed the Wilcoxon test with a *p*-value of <0.05 as statistically significant. The Bland–Altman analysis and intraclass correlation coefficients (ICC) were employed to evaluate the agreement between package A and B as well as agreement between both packages and follow-up DWI. ICC were interpreted as proposed: <0.50 (poor), 0.50–0.75 (moderate), 0.75–0.90 (good) and >0.90 (excellent).

Neuroimaging eligibility criteria included in DEFUSE3 were applied for individual patient triage to determine the concordance of treatment decisions based on two packages. Specifically, mismatched profiles, including mismatch volume (i.e., the difference in volumes between penumbra and ischemic core) and mismatch ratio (i.e., the ratio of penumbra and ischemic core), were calculated accordingly using volumetric results. Then, the eligibility of MT for individual AIS patients was assessed derived from each package, and the agreement of patient triage was measured by ICC. To differentiate the factor being either overwhelmed ICV or malignant mismatch profiles that caused the disagreement, patients were divided into subgroups according to package A-based ICV with a threshold of 70 mL. In addition, because the mismatch profiles were used as criteria to triage patients and the differences in mismatch values were not the focus of this study, we did not compare the computing results.

## 3. Results

### 3.1. Patient Characteristics

The clinical characteristics of patients recruited in this study were summarized in Table 1. CTP taken by 174 patients (mean age: 65 years) and MRI taken by 420 patients (mean age: 64 years) were used.

### 3.2. Comparison of Measurements between Packages

Comparison of ischemic core volume (ICV) and penumbra volume (PV) in CTP group with ICV > 70 mL and ICV < 70 mL between two packages is shown in Figure 3 and Figure 4, respectively. Comparison of ischemic core volume (ICV) and penumbra volume (PV) in MRI group with ICV > 70 mL and ICV < 70 mL between two packages is shown in Figure 5 and Figure 6, respectively.

The mean (SD) and median values (IQR) for ICV and PV are summarized in Table 2. In CTP scans, mean ICVs were 14.9 (36.0) mL and 14.0 (28.3) mL; mean PVs were 81.1 (95.7) mL and 83.2 (91.4) mL for package A and B, respectively. The Wilcoxon test revealed no significant difference in ICV (*p* = 0.264) and PV (*p* = 0.354) measurements. As to the 420 MRI scans, mean ICVs were 52.4 (69.5) mL and 48.9 (69.1) mL; mean PVs were 68.4 (77.3) mL and 61.6 (72.6) mL for package A and B, respectively, and the measurements did not differ significantly (ICV: *p* = 0.463 and PV: *p* = 0.178).

As shown in Table 3, volumetric results derived from the two packages showed strong agreement regardless of imaging modality. In the CTP group, the mean differences were 0.89 (12.7) and −2.0 (13.0) mL for ICV and PV, respectively; in the MRI group, the mean differences were 3.5 (4.1) and 10.8 (46.9) mL for ICV and PV, respectively. Package B showed excellent agreement with package A in estimating ICV in both CTP and MRI groups with ICCs above 0.95. Good agreement was found in PV measurement in the MRI group with a slightly lower ICC of 0.87.

The Bland–Altman analysis showed the measuring difference in relation to the volume. As shown in Figure 7 and Figure 8, in the CTP group, positive bias was found in ICV measurement. The bias seemed to be mainly driven by measurements with a mean volume above 70 mL, leading to an underestimated result for large ischemic core by package B. PV measurement, however, showed a negative bias, and this was possibly attributed to the overestimation by package B for objects with size below 100 mL. In the MRI group, both ICV and PV showed positive biases between the two packages. This might be caused by the underestimation by package B at a volume greater than 50 mL, indicating a better agreement between packages in measuring small objects.

In order to validate the accuracy of CTP ICV, this study included follow-up DWI with manual lesion labeling as ground truth of ischemic core to compare the CTP performance of package A against package B. Among the 174 AIS patients with CTP scanning, 53 patients had follow-up DWI. This study calculated the volume of labeled AIS lesions and compared the CTP ICV derived from package A and B with the follow-up DWI lesion volume. As shown in Table 4, volumetric results derived from the two packages showed strong agreement with follow-up DWI. The mean differences were −4.65 (16.3) mL and −3.65 (16.3) mL for package A and B, respectively. Both packages showed excellent volumetric agreement with follow-up DWI in estimating CTP ICV. The ICC of package A was 0.92, while the ICC of package B was 0.94. The CTP ICV derived from package B was more accurate than package A in comparison with the follow-up DWI lesion volume.

The Bland–Altman analysis showed the measuring difference in relation to the volume. As shown in Figure 9 and Figure 10, negative bias was found in CTP ICV measurement for both packages. The bias seemed to be mainly driven by measurements with a mean volume below 10 mL, leading to an underestimated result for small ischemic core. In 12 AIS cases, CTP ICV and follow-up DWI lesion volume were below 3 mL. In eight AIS cases, CTP ICV was below 3 mL while follow-up DWI lesion volume was 3–10 mL. In two AIS cases, follow-up DWI lesion volume was below 3 mL while CTP ICV was 3–10 mL.

### 3.3. Comparison of Patient Triage

Diagnostic agreement based on DEFUSE3 criteria between packages was analyzed, and the number of patients eligible for MT is listed in Table 5. Of 174 CTP scans, excellent agreement was achieved with ICC of 0.90. Concordance was found in 16/19 (84%) cases in subgroup with package-A-based ICV > 70 mL, and 143/155 case (92%) in the subgroup with ICV < 70 mL. Of 420 MRI scans, excellent agreement was achieved with ICC of 0.93. Concordant results were obtained in 125/130 (96%) in the subgroup with ICV > 70 mL and 275/290 (94%) in the subgroup with ICV < 70 mL. Interestingly, in subgroups with a large ischemic core, eight cases with discrepant treatment decisions from CTP and MRI group had smaller ICV based on package B, qualifying patients for MT, while package A did not qualify. In subgroups with ICV below the threshold of 70 mL, different biases in ICV and PV led to discordance in mismatched profiles, which affected patient selection for MT. Specifically, 15 patients were assumed to be ineligible for MT by package A solely, with 9 of them due to an inadequate mismatch ratio and 4 of them due to inadequate mismatch volume; while 12 patients were excluded from MT by package B with only 8 patients due to a malignant mismatch ratio and 4 patients due to a malignant mismatch volume.

## 4. Discussion

This study evaluated the agreement between two software packages for both CTP and MR postprocessing modules. Our findings suggested there was a remarkable concordance between the performances of the two packages in terms of estimating ischemic core and penumbra volumes as well as making treatment decisions for AIS patients. Compared with package A, package B classified more patients as candidacies for MT.

Accurate estimation of ICV and PV is of great importance. The under- and overestimation of volumes may have important consequences in clinical practice [10]. With a cutoff ICV of 70 mL being proposed as one of the inclusion criteria in DEFUSE3, overestimating ICV may inaccurately exclude patients from MT, while underestimating ICV may result in poor clinical outcomes due to hemorrhage following endovascular procedures [14]. As for PV estimation, since the volume indicates the amount of potentially salvageable tissue by timely treatment [15], accurate measurement is crucial to finding patients who will respond well to MT [16]. Previous software comparison studies found discrepant volumetric results among different software packages [17,18,19,20]. A software comparison study found that IntelliSpace Portal CT Brain Perfusion (Royal Philips Healthcare, Best, The Netherlands) and syngo.via (Siemens Healthcare, Erlange, Germany) showed increased deviation from Rapid, with ICV increasing from 25 mL to over 70 mL [10]. The discrepancy is related to the differences in postprocessing algorithms such as defining arterial input, motion correction and smoothing [1,21,22]. The results of this study were in line with those of previous studies, where variation in volumetric measurement in relation to the size were described.

Compared with package A, package B underestimated volumes in both groups except CTP penumbra. Previous studies indicated that penumbra was likely to be overestimated in package A [14,23,24,25]. Researchers attributed this to several factors: (1) artificial volumes for the sake of presenting, (2) erroneous T_max_ calculations with some cases showing bi-hemispheric penumbra due to the demyelination of bilateral periventricular white matter, which indicated myelin sheath injury mediated by cerebral small vessel diseases [26], and (3) altered arterial input function attributed to quantum noise during imaging acquisition. Given these facts, the underestimation of PV in package B seems preferable. Moreover, cases with inappropriate bi-hemispheric penumbra were fewer in results derived from package B. Additional restrictions may be applied in the postprocessing algorithm in package B to avoid unreasonable outcomes.

For the purpose of validating the accuracy of CTP ICV, 24 h follow-up DWI was included and labeled as ground truth. DWI is a useful method to detect diffusivity capacity of water molecule with high sensitivity, especially for AIS, with cytotoxic edema showing a high signal on DWI. The DWI-restricted lesions were once regarded as representation of the damaged ischemic core. However, DWI lesion reversal occurred in some patients with ischemic stroke indicating the alleviation of cytotoxic edema. A systematic review of the published literature on DWI hyperintense tissue outcome reported variable rates of DWI reversal (0–83%), with a mean reversal rate of 24% in patients with ischemic stroke [27]. In a previous study, DWI reversal was associated with small infarct volume in patients with TIA (Transient Ischemic Attacks) and minor stroke [28]. In our study, two minor AIS cases with CTP ICV 3–10 mL and follow-up DWI lesion volume below 3 mL may indicate DWI reversal caused by timely reperfusion of ischemic region occurred in these AIS cases. Eight AIS cases with CTP ICV below 3 mL and follow-up DWI lesion volume 3–10 mL may be attributed to the small amount of cytotoxic edema existing around acute infarct, which was not sensitive for CT to detect tiny HU changes caused by cytotoxic edema. Twelve AIS cases with CTP ICV and follow-up DWI lesion volume below 3 mL may be attributed to good collateral circulation around the infarct. Furthermore, since there is no independent imaging approach for verifying penumbra measurement [29], the reference standard is lacking for assessing which package is more accurate.

As comparing differences in volume independently cannot explain the impact of these changes in a clinical setting, we moved forward to assess the agreement between packages in patient selection for MT. It is verified that treatment selections could vary at the individual level despite similar volumetric results between packages. Simultaneously, patient triage may rarely be affected even though volumetric estimations were of apparent discordance [23]. For instance, one patient will be excluded from MT based on the result of package A (ICV/PV: 74/164 mL), but the result measured by package B (ICV/PV: 67/148 mL) will lead the other way. Inappropriate patient stratification is likely to increase the risk of undesirable clinical outcomes [30]. When evaluating the performance of these software packages, the accuracy of triaging AIS patients based on neuroimaging inclusion criteria needs to be considered.

This study has several limitations. First, a small sample size for CTP limited the power to detect differences between software packages. Only a small number of cases had package-A-based ICV larger than 70 mL (*n* = 19), limiting the power to accurately test the consistency of patient triage between packages. Second, CTP data acquired in this study were obtained from two different scanner platforms, and MR data were from only one scanner platform. Data from various scanner brands are preferred to satisfy data heterogeneity. Furthermore, one exclusion criterion for imaging data relied on the success of postprocessing in package A, and some CTP cases were excluded because of the processing errors. These failures in package A can be attributed to poor image quality or the way of managing time-series data. With this restriction, data variability was affected, and the postprocessing results of abnormal cases were not considered in this study. Another limitation is the lack of enough cases of large ischemic core shown on 24 h follow-up MRI performed after CTP examination. DWI data are reliable for identifying ischemic core [31,32], and follow-up DWI that were acquired within the recommended time window have been applied in prior studies for determining the accuracy of volumetric and spatial estimation of ischemic core on CTP [10,33]. However, 24 h follow-up MRI was not available for some severe AIS patients in this study because of the postponement and contraindications of follow-up MR examinations. Additionally, 24 h follow-up MRI was not a clinical routine for AIS patients who had accepted DWI-PWI examination. Last, the final ischemic core was no longer comparable due to the possible penumbra evolution [34]. Considering that RAPID-based results have been validated in previous studies [11,35,36,37,38], using results derived from RAPID as reference remains clinically relevant.

## 5. Conclusions

RealNow has similar volumetric measurements as RAPID for CTP and MRI postpossessing modules. A high degree of concordance between the two packages was achieved in terms of making treatment decisions for AIS patients. Our study suggested that RAPID and RealNow can be used interchangeably. As for CTP ICV, RealNow was validated to be more accurate than RAPID by 24 h follow-up DWI. Since we did not validate DWI and PWI for each method separately, we refrain from concluding which package is more accurate in MRI. Additional research is required to validate the accuracy of volumetric measurements and patient triage.

## Figures and Tables

**Figure 1 cells-11-02547-f001:**
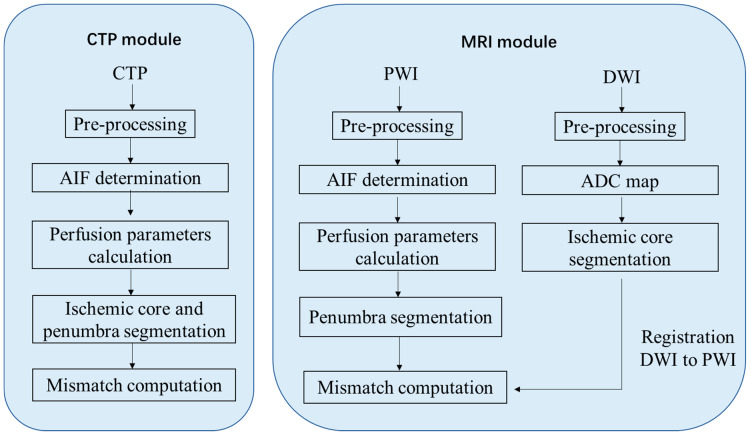
Postprocessing procedure for perfusion imaging.

**Figure 2 cells-11-02547-f002:**
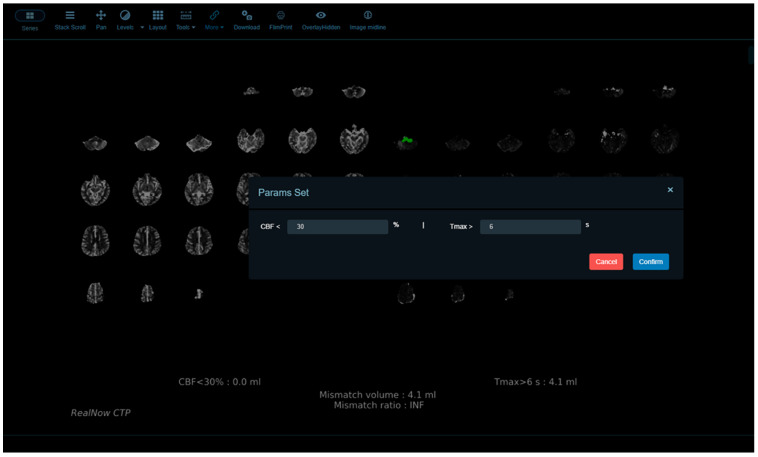
Package B provides a user interface online that allows the thresholds for rCBF, ADC and T_max_ to be self-defined.

**Figure 3 cells-11-02547-f003:**
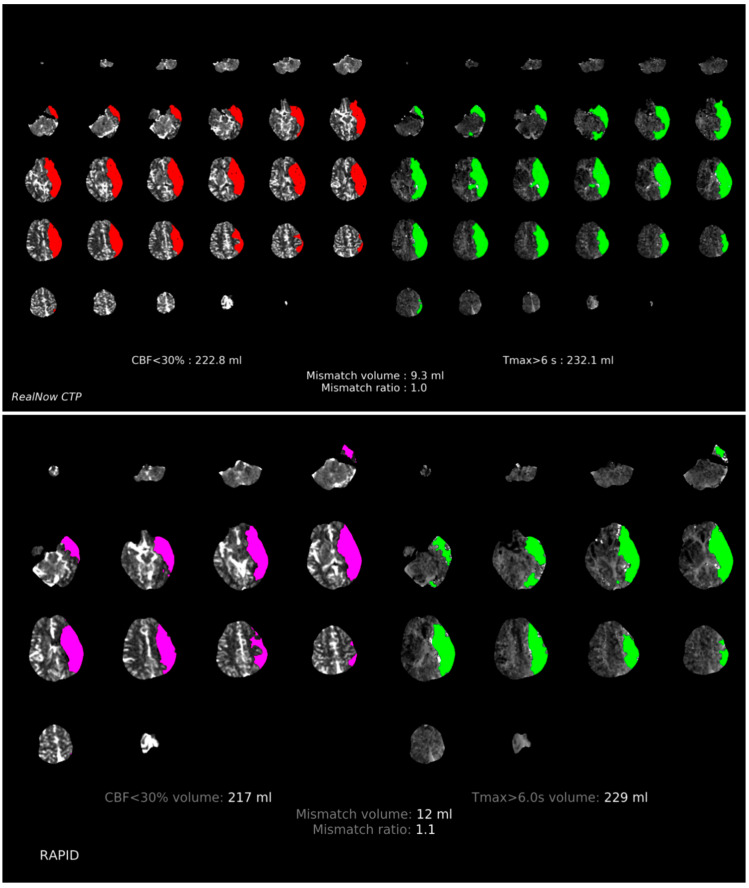
Comparison of ischemic core volume (ICV) and penumbra volume (PV) in CTP group with ICV > 70 mL between two packages. Red and fuchsia areas are ischemic core defined by package A and B, respectively. Green areas are penumbra.

**Figure 4 cells-11-02547-f004:**
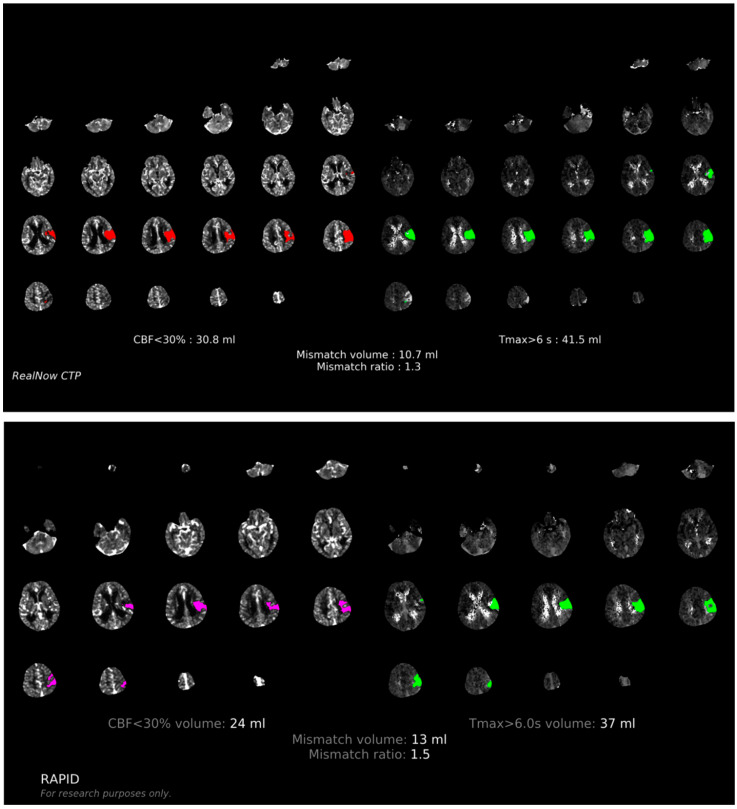
Comparison of ischemic core volume (ICV) and penumbra volume (PV) in CTP group with ICV < 70 mL between two packages. Red and fuchsia areas are ischemic core defined by package A and B, respectively. Green areas are penumbra.

**Figure 5 cells-11-02547-f005:**
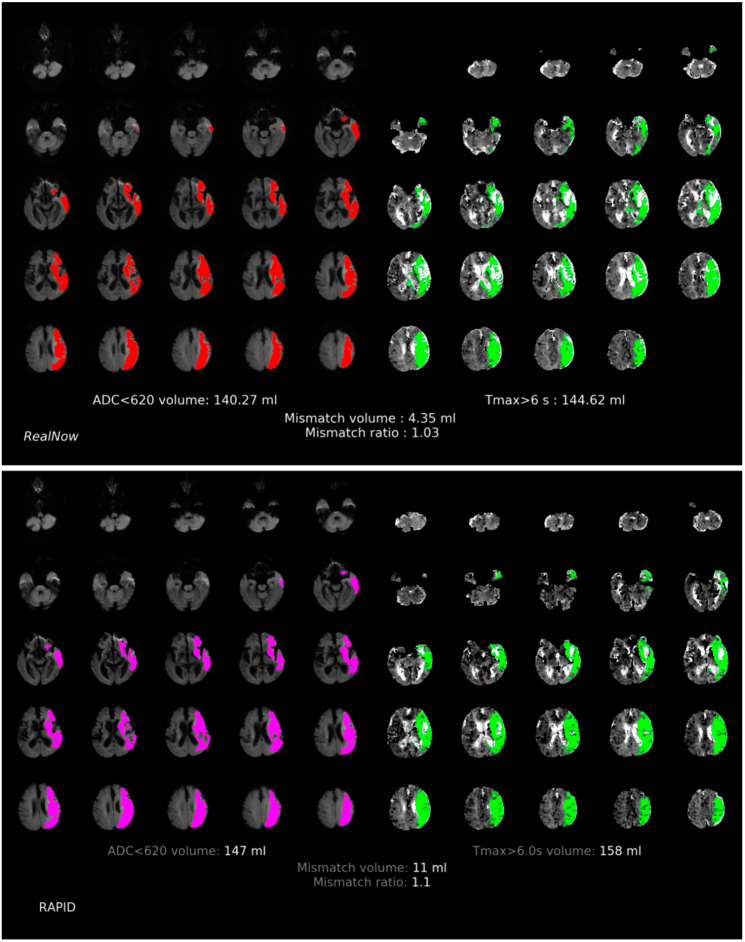
Comparison of ischemic core volume (ICV) and penumbra volume (PV) in MRI group with ICV > 70 mL between two packages. Red and fuchsia areas are ischemic core defined by package A and B, respectively. Green areas are penumbra.

**Figure 6 cells-11-02547-f006:**
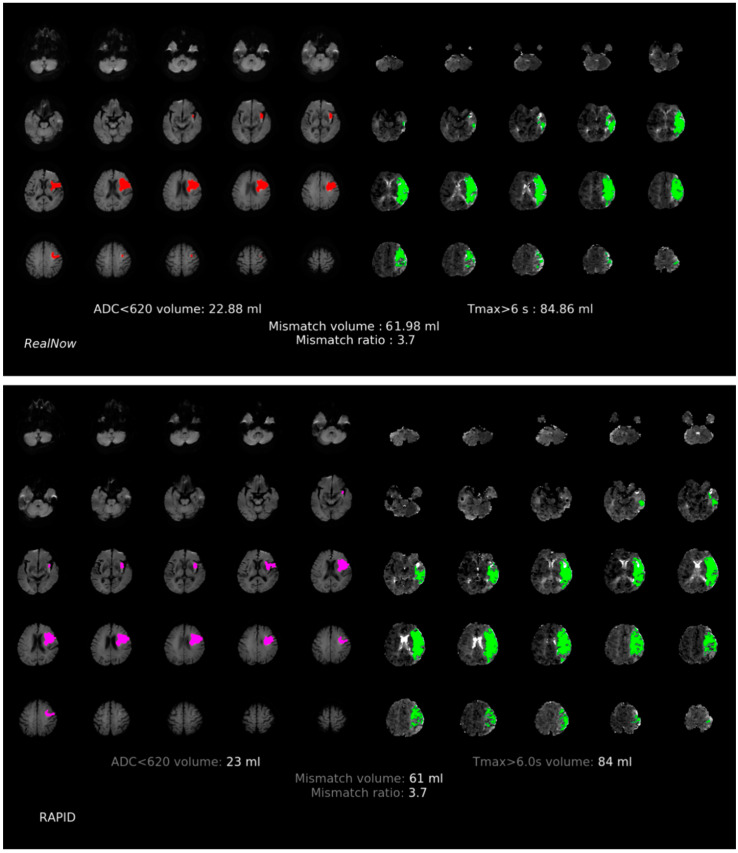
Comparison of ischemic core volume (ICV) and penumbra volume (PV) in MRI group with ICV < 70 mL between two packages. Red and fuchsia areas are ischemic core defined by package A and B, respectively. Green areas are penumbra.

**Figure 7 cells-11-02547-f007:**
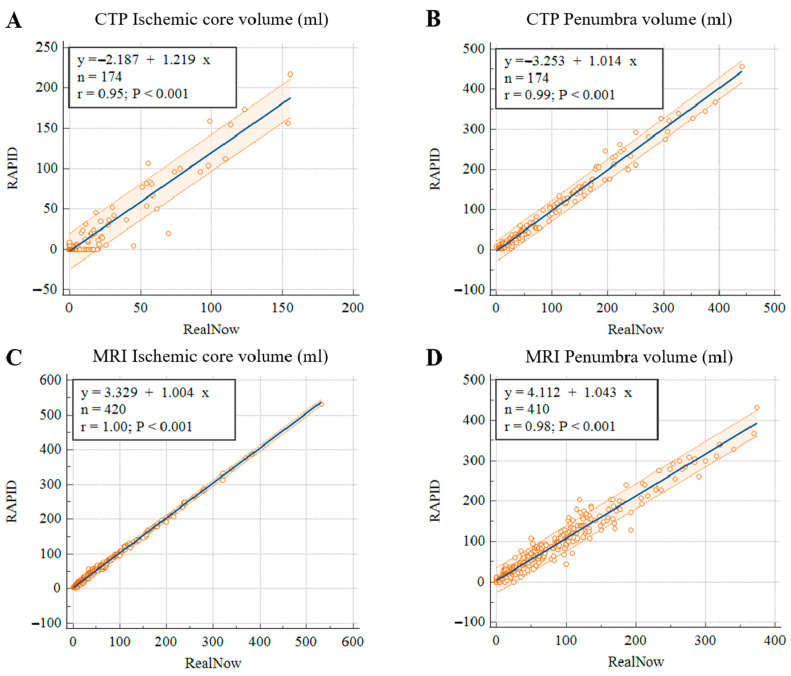
Scatter plots of ischemic core (**A**) and penumbra volumes (**B**) in CTP group, ischemic core (**C**) and penumbra volumes (**D**) in MRI group.

**Figure 8 cells-11-02547-f008:**
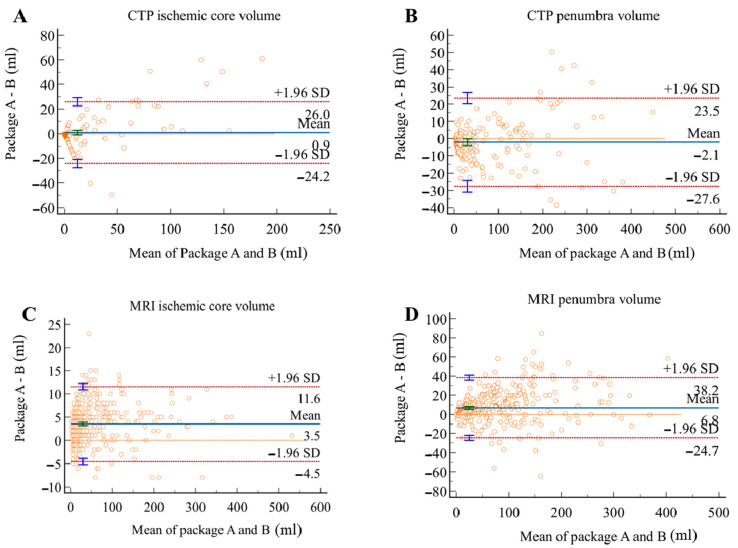
Bland–Altman plots for volumetric agreement of ischemic core (**A**) and penumbra volumes (**B**) in CTP group, ischemic core (**C**) and penumbra volumes (**D**) in MRI group. Solid lines represent the mean difference between two packages. Dotted lines represent 95% limits of agreement.

**Figure 9 cells-11-02547-f009:**
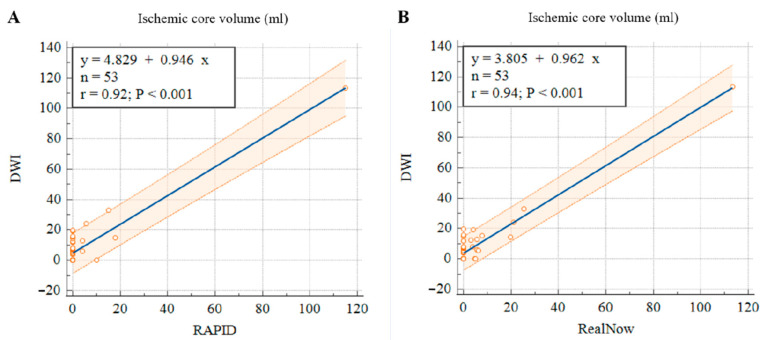
Scatter plots of ischemic core volumes estimated with RAPID versus follow-up DWI (**A**) and RealNow versus follow-up DWI (**B**).

**Figure 10 cells-11-02547-f010:**
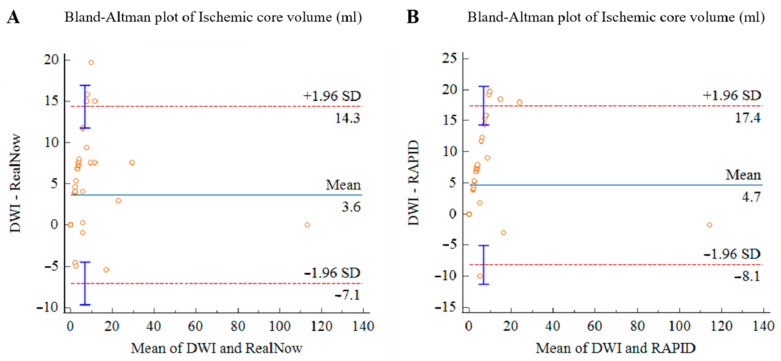
Bland–Altman plots for volumetric agreement of ischemic core estimated with RAPID versus follow-up DWI (**A**) and RealNow versus follow-up DWI (**B**). Solid lines represent the mean difference between two packages. Dotted lines represent 95% limits of agreement.

**Table 1 cells-11-02547-t001:** Demography of subjects in two groups.

Groups	CTP Group (*n* = 174)	MRI Group (*n* = 420)
Age in years (SD)	65 (20.6)	64 (17.1)
Gender(M/F)	135/39	286/134
Occlusion sites, *n* (%)		
Middle cerebral artery	101 (58.0)	218 (51.9)
Posterior cerebral artery	17 (9.8)	58(13.8)
Anterior cerebral artery	11 (6.3)	32 (7.6)
Internal carotid artery	33 (18.9)	74 (17.6)
Basilar artery	12 (6.8)	38(9.0)

SD, standard deviation.

**Table 2 cells-11-02547-t002:** Comparison of ischemic core volume (ICV) and penumbra volume (PV) between two packages.

			Package A	Package B
CTP group	ICV, mL	Mean (SD)	14.9 (36.0)	14.0 (28.3)
		median [IQR]	0.0 [0.0–6.4]	2.7 [0.0–13.1]
	PV, mL	Mean (SD)	81.1 (95.7)	83.2 (91.4)
		median [IQR]	39.2 [12.0–129.1]	43.7 [14.6–128.3]
MRI group	ICV, mL	Mean (SD)	52.4 (69.5)	48.9 (69.1)
		median [IQR]	24.0 [12.0–64.0]	21 [0.0–59.5]
	PV, mL	Mean (SD)	68.4 (77.3)	61.6 (72.6)
		median [IQR]	44.0 [4.0–108.0]	38.3 [0.0–99.4]

ICV, ischemic core stroke volume; PV, penumbra volume.

**Table 3 cells-11-02547-t003:** Mean difference, limits of agreement (LoA), intraclass correlation coefficient (ICC) and Wilcoxon test between two packages.

	CTP Group		MRI Group	
	ICV	PV	ICV	PV
Mean difference (SD), mL	0.89 (12.7)	−2.0 (13.0)	3.5 (4.1)	6.8 (46.9)
95% Lower LoA (95% CI)	−24.1 (−27.4 to −20.9)	−27.6 (−30.9 to −24.3)	−4.5 (−5.1 to −3.8)	−81.2 (−89.6 to −72.8)
95% Upperer LoA (95% CI)	25.9 (22.6 to 29.2)	23.4 (20.1 to 26.8)	11.5 (10.8 to 12.2)	102.9 (94.6 to 111.3)
ICC (95% CI)	0.95 (0.94 to 0.97)	0.99 (0.98 to 0.99)	0.99 (0.98 to 0.99)	0.87 (0.84 to 0.89)
Wilcoxon test (*p* value)	0.264	0.354	0.463	0.178

Package A as reference; package B as variable; difference: reference—variable.

**Table 4 cells-11-02547-t004:** Mean difference, limits of agreement (LoA), intraclass correlation coefficient (ICC) and Wilcoxon test between two packages and follow-up DWI.

	RAPID	RealNow
Mean difference (SD), mL	−4.65 (16.3)	−3.65 (16.3)
95% Lower LoA (95% CI)	−8.14 (−11.2 to −5.05)	−7.06 (−9.64 to −4.47)
95% Upperer LoA (95% CI)	17.4 (14.4 to 20.5)	14.3 (11.8 to 16.9)
ICC (95% CI)	0.92 (0.86 to 0.95)	0.94 (0.90 to 0.97)
Wilcoxon test (*p* value)	0.144	0.253

Follow-up DWI as reference; package A and B as variable; difference: reference—variable.

**Table 5 cells-11-02547-t005:** Patient triage for mechanical thrombectomy (MT) conducted by two packages based on imaging eligibility criteria included in DEFUSE3.

	ICC	Subgroup	Package A		Package B	
			Eligible	Not Eligible	Eligible	Not Eligible
CTP group	0.90	ICV > 70 mL	16	3	19	0
		ICV < 70 mL	145	10	153	2
MRI group	0.93	ICV > 70 mL	125	5	130	0
		ICV < 70 mL	285	5	280	10

## Data Availability

Data may be made available from the corresponding authors upon reasonable request.

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
