# Peer review of "Comparison of Two Software Packages for Perfusion Imaging: Ischemic Core and Penumbra Estimation and Patient Triage in Acute Ischemic Stroke"

_cells, 2022, doi:10.3390/cells11162547_

Round 1

Reviewer 1 Report

In this study, a comparison of two software packages for perfusion imaging: ischemic core and penumbra estimation and patient triage in acute ischemic stroke has been performed. From algorithm side, there is no novelty, because the authors “just” used the existing software packages. However, the authors evaluated a total of 594 AIS patients, which is quite comprehensive. I find this work quite important and more works should evaluate existing software packages in medical image processing to see how they really perform.

Author Response

Thank you for your valuable advice. Based on your comment and request, we have made extensive modifications on the original manuscript, and the modifications have been highlighted for easy check/editing purpose. We have added introduction to RealNow software processing of CTP and MR images in details as supplementary, which can help readers better understand the processing stages of RealNow. Here, we have attached revised manuscript in the formats of MS word, for your approval. In addition, based on the existing research results, we will recruit more AIS patients with CTP and MRI to evaluate different software packages in medical image processing and compare their clinical application value in AIS diagnosis and triage.

Reviewer 2 Report

1. The significance of the study is not clear. How it is necessary to compare the two software packages?

2. Pre_processing: IS the pre-processing of the images is performed by authors or its performed by the software packages? 

3. What do the authors mean by the self-defined thresholds? how is this defined and the criterion?

4. Figure 4-6; define the color-coding?

5.  From the Bland Altman plot it is clear that the concentration is not along with the mean. please justify.

6. It is not clear how the study is concluded about recommending one package which may lead to favouring  a method.

Author Response

  1. The significance of the study is not clear. How it is necessary to compare the two software packages?

Reply: As mentioned in line 56-64: ‘…different biases in volumes of ischemic core and penumbra can act associatively and affect patient triage differently…’. Previous studies compared the differences between software packages, and they mostly focused on comparing ischemic core and penumbra volumes. However, none of these studies take mismatch profile into account. Since mismatch profile influences mechanical thrombectomy outcomes and treatment decisions are made based on mismatch profile, this manuscript incorporates comparison of the patient triage between two software packages. In clinical setting, accurate and timely patient triage is essential to patients with acute ischemic stroke. The concordance of the patient triage results provided by different vendors needs to be evaluated. With computer-aided software packages providing identical volumetric measurements and patient triage results, radiologists can be more confident of making clinical decision.

  1. Pre-processing: IS the pre-processing of the images is performed by authors or its performed by the software packages?

Reply: Pre-processing is performed by the packages. We provide a supplementary material describing the embedded algorithms applied in RealNow. The pre-processing includes motion correction, spatial smoothing and brain extraction.

  1. What do the authors mean by the self-defined thresholds? how is this defined and the criterion?

Reply: The thresholds rCBF, ADC and Tmax can be self-defined in package B. To do this, users need to input numeric values in the dialog (see Figure 2). The reasons for allowing self-defining thresholds are 1) the default thresholds may not apply in all circumstances and 2) it helps researchers investigate the optimal thresholds for defining lesions for patients with different causes or time of onset. We addressed this question in line 114-116, and 121-123.

  1. Figure 4-6; define the color-coding?

Reply: The definition is added in the titles of each figure for the new version of manuscript. Red and fuchsia areas are ischemic core defined by package A and B, respectively. Green areas are penumbra defined by package A and B.

  1. From the Bland Altman plot it is clear that the concentration is not along with the mean. please justify.

Reply: We justified this issue in line 253-260. As shown in Figure 8, the volumetric difference between RealNow and Rapid increased along with the increase of ICV or PV. In the CTP group, positive bias was found in ICV measurement. The bias seemed to be mainly driven by measurements with a mean volume above 70 ml, leading to an underestimated result for large ischemic core by package B. PV measurement, however, showed a negative bias, and this was possibly attributed to the overestimation by package B for objects with size below 100 ml. In the MRI group, both ICV and PV showed positive biases between the two packages. This might be caused by the underestimation by package B at a volume greater than 50 ml, indicating a better agreement between packages in measuring small objects. In addition, a software comparison study also found that IntelliSpace Portal CT Brain Perfusion (Royal Philips Healthcare, Best, The Netherlands) and syngo.via (Siemens Healthcare, Erlange, Germany) showed increased deviation from Rapid with ICV increasing from 25 ml to over 70 ml. The discrepancy is related to the differences in post-processing algorithms such as defining arterial input, motion correction and smoothing. The results of this study were in line with those of previous studies, where variation in volumetric measurement in relation to the size were described.

  1. It is not clear how the study is concluded about recommending one package which may lead to favouring a method.

Reply: In this study, we included follow-up DWI with manual lesion labeling as ground truth of ischemic core to compare the CTP performance of RealNow against RAPID package. Among the 174 AIS patients with CTP scanning, 53 patients has 24-hour follow-up DWI. The boundary of AIS lesions with high signal on DWI were manually delineated. This study calculated the volume of labeled AIS lesions and compared the CTP ICV derived from RealNow and RAPID package with the follow-up DWI lesion volume. As shown in Table 4, both software packages showed excellent volumetric agreement with follow-up DWI in estimating CTP ICV. The ICC of Rapid was 0.92, while the ICC of RealNow was 0.94. The CTP ICV derived from RealNow was more accurate than Rapid in comparison of follow-up DWI lesion volume. Therefore, we favoured RealNow for CTP ICV’s calculation.

Reviewer 3 Report

 Zhou and colleagues performed a comparison of 2 software packages devoted to quantify ICV and PV in both CT and MRI (perfusion and diffusion-weighted images). The aim of such a study is validate their homemade tool against a state-of-art FDA-approved software. This is a nice study, whereas I have some concerns that must be addressed before the publication.

Major:

1. Since I did not find a previous publication about the RealNow software, the authors should add a comprehensive section describing how the image processing is done. This is very important since the authors are validating their method against a state-of-art tool. The more details they provide, better will be.

2. The CTP data were acquired in two different CT scanners. Were the protocols harmonized? The parameters provided must be improved, especially to the MRI protocol. How many dynamics were acquired to the PWI?

3. Again, the authors must provide more details about how they calculate the penumbra volume e ischemic core. Otherwise, it will be very difficult to understand the reliability and reproducibility of the RealNow.

4. In the statistical analysis, ICC is not the most appropriate method to quantify the agreement between the methods. ICC is better used to assess reproducibility of tool in a test-retest dataset, i.e., it provides the agreement of repeated measures for a same tool. The authors are comparing measures from two different softwares. I suggest to use Dice score and Hausdorff distance.

5. In order to better investigate and validate the RealNow software, I suggest to perform ROC curves to compare the performance of RealNow against RAPID package. The authors might use the DEFUSE3 dataset to accomplish that.

Author Response

  1. Since I did not find a previous publication about the RealNow software, the authors should add a comprehensive section describing how the image processing is done. This is very important since the authors are validating their method against a state-of-art tool. The more details they provide, better will be.

Reply: According to the valuable suggestion, describing algorithm enable researchers understand the differences between software packages. Thus, we submit a supplementary document describing the detailed processing methods used in RealNow, which can help readers better understand the processing stages of RealNow.

  1. The CTP data were acquired in two different CT scanners. Were the protocols harmonized? The parameters provided must be improved, especially to the MRI protocol. How many dynamics were acquired to the PWI?

Reply: According to the reviewer’s valuable advise, more detailed CTP and MRI scanning parameters have been added into the Tabel S1 and S2, such as 60 dynamics acquired to the PWI. This study was a multicenter and retrospective study, while the scanning parameters of SIEMENS and TOSHIBA CTP protocols used by three hospitals were partly different. The optimal CTP scanning parameters were adjusted according to specific CT scanners by the engineers of SIEMENS and TOSHIBA for the balance of imaging quality, scanning speed and radiation dose. The SOMATOM Definition Flash of SIEMENS contained two X-ray tubes with two 64-row detectors. And Aquilion ONE of TOSHIBA contained One X-ray tube with one 320-row detector. The scanning speed, coverage and spacial resolution of both CT scanners were different. Therefore, some scanning parameters such as the brain coverage, mAs, and number of scanning phases of both CT scanners were not the same. The CT images derived from both CTP protocols met the requirements of CTP calculation and diagnosis. The SIEMENS and TOSHIBA CTP protocols were also used for CTP imaging acquisition and post-processing in other researches and clinical use.

  1. Again, the authors must provide more details about how they calculate the penumbra volume & ischemic core. Otherwise, it will be very difficult to understand the reliability and reproducibility of the RealNow.

Reply: This question is similar to question 1. We have provided a supplementary material to elaborate the post-processing methods, which can help readers better understand the reliability and reproducibility of RealNow.

  1. In the statistical analysis, ICC is not the most appropriate method to quantify the agreement between the methods. ICC is better used to assess reproducibility of tool in a test-retest dataset, i.e., it provides the agreement of repeated measures for a same tool. The authors are comparing measures from two different softwares. I suggest to use Dice score and Hausdorff distance.

Reply: This is a very good question. To calculate dice score and Hausdorff distance, we need to compare the NII formatted segmentation mask at the same location (i.e., the location of a voxel in a 3D volume). However, since Rapid software provides summary calculation results of CTP and MRI slices in a picture format (Figure 3-6) , while Rapid cannot provide NII formatted lesion segmentation mask of each slice, the comparison of dice score and Hausdorff distance between RealNow and Rapid software is limited. In addition, the previous software comparison study used Bland-Altman plots for volumetric agreement of core volumes derived from different software. The study found that IntelliSpace Portal CT Brain Perfusion (Royal Philips Healthcare, Best, The Netherlands) and syngo.via (Siemens Healthcare, Erlange, Germany) showed increased deviation from Rapid with ICV increasing from 25 ml to over 70 ml. Our results were in line with previous studies, where variation in volumetric measurement in relation to the size were described.

  1. In order to better investigate and validate the RealNow software, I suggest to perform ROC curves to compare the performance of RealNow against RAPID package. The authors might use the DEFUSE3 dataset to accomplish that.

Reply: Thank you for arising a valuable question. Previous studies used Bland-Altman analysis and intraclass correlation coefficients (ICC) to evaluate the volumetric agreement between two software packages. A Previous study used Bland-Altman plots for volumetric agreement of core volumes with ISP, syngo.via methods A, B, and C and RAPID and found poor agreement between RAPID and ISP software using default thresholds. Moreover, a previous study analyzed agreement of detecting ischemic core volumes with thresholds of ≤25mL, ≤50mL and ≤70 mL without using ROC curves. ROC curves has been used to determine the optimal threshold of ischemic core and penumbra qualified for mechanical thrombectomy (MT) in previous clinical trails. A cutoff ICV of 70 ml was proposed as one of the inclusion criteria to select patients for MT in DEFUSE3. We did not explore triage standards for MT selection using ROC curves in this study. If we used ROC curves to compare the performance of RealNow against RAPID package, it would focus on the seeking the optimal CTP or MRI threshold for AIS diagnosis or classification according to the largest AUC (areas under the curves), and then diagnosis or triage accuracy of CTP or MRI threshold derived from two software packages would be compared. Therefore, we will conduct a prospective study on the optimal triage threshold for MT selection based on infarct core volume, penumbra volume and mismatch ratio calculated by different software to compare the clinical application value of the threshold by using ROC curves analyses in the near future.

Since the DEFUSE3 data set is non-public, we have not been authorized to use this data set for further verification. In this study, we included follow-up DWI with manual lesion labeling as ground truth of ischemic core to compare the CTP performance of RealNow against RAPID package. Among the 174 AIS patients with CTP scanning, 53 patients has 24-hour follow-up DWI. The boundary of AIS lesions with high signal on DWI were manually delineated. This study calculated the volume of labeled AIS lesions and compared the CTP ICV derived from RealNow and RAPID package with the follow-up DWI lesion volume. As shown in Table 4, both software packages showed excellent volumetric agreement with follow-up DWI in estimating CTP ICV. The ICC of Rapid was 0.92, while the ICC of RealNow was 0.94. The CTP ICV derived from RealNow was more accurate than Rapid in comparison of follow-up DWI lesion volume.

Reviewer 4 Report

This paper compares the results of  the application of two software packages for the analysis of computed tomographic perfusion imaging (CTP) data or magnetic resonance imaging (MRI) data in the acute ischemic stroke patients.

The conclusions of the research are rather confusing. On the one hand, the authors state that both packages produced similar volumetric measurements of the ischemic core , and penumbra volumes both for CTP and MRI. On the other hand, the results did not allow for any conclusion on  which package is more accurate, and further research is required to figure that out. I believe that such conclusions provide no helpful information on which package to use in which case (CTP or MRI). Furthermore,  the research topic of this paper does appear to belong to the scope of the journal.

Author Response

Thank you for your valuable advice. Based on your comment and request, we have made extensive modifications on the original manuscript, and the modifications have been highlighted for easy check/editing purpose. Here, we have attached revised manuscript in the formats of MS word, for your approval. 

In this study, we included follow-up DWI with manual lesion labeling as ground truth of ischemic core to compare the CTP performance of RealNow against RAPID package. Among the 174 AIS patients with CTP scanning, 53 patients has 24-hour follow-up DWI. The boundary of AIS lesions with high signal on DWI were manually delineated. This study calculated the volume of labeled AIS lesions and compared the CTP ICV derived from RealNow and RAPID package with the follow-up DWI lesion volume. As shown in Table 4, both software packages showed excellent volumetric agreement with follow-up DWI in estimating CTP ICV. The ICC of Rapid was 0.92, while the ICC of RealNow was 0.94. The CTP ICV derived from RealNow was more accurate than Rapid in comparison of follow-up DWI lesion volume. Therefore, we favoured RealNow for CTP ICV’s calculation.

Futhermore, we have added introduction to RealNow software processing of CTP and MR images in details as supplementary, which can help readers better understand the processing stages of RealNow. 

We will recruit more AIS patients with CTP and MRI to evaluate different software packages in medical image processing and to compare their clinical application value in AIS diagnosis and triage.

Round 2

Reviewer 3 Report

The authors addressed all my comments.